# The Bamboo Rhizome Evolution in China Is Driven by Geographical Isolation and Trait Differentiation

Han-Jiao Gu [1,2,†], Can-Can Zhang [1,†], Fu-Sheng Chen [1,3], Ji-Hong Huang [4], Jin-Song Wang [5], Helge Bruelheide [6,7], Stefan Trogisch [6,7], Xiang-Min Fang [1], Jian-Jun Li [1] and Wen-Sheng Bu [1,3,6,*,‡]

1   Key Laboratory of State Forestry Administration on Forest Ecosystem Protection and Restoration of Poyang Lake Watershed, College of Forestry, Jiangxi Agricultural University, Nanchang 330045, China; guhanjiao@163.com (H.-J.G.); brightzcc@163.com (C.-C.Z.); chenfush@yahoo.com (F.-S.C.); xmfang2013@126.com (X.-M.F.); lijianjun8801@126.com (J.-J.L.)
2   State Key Laboratory of Vegetation and Environmental Change, Institute of Botany, Chinese Academy of Sciences, Beijing 100093, China
3   Jiulianshan National Observation and Research Station of Chinese Forest Ecosystem, Jiangxi Agricultural University, Nanchang 330045, China
4   Key Laboratory of Forest Ecology and Environment of State Forestry Administration, Institute of Forest Ecology, Environment and Protection, Chinese Academy of Forestry, Beijing 100091, China; northalluvion@caf.ac.cn
5   Key Laboratory of Ecosystem Network Observation and Modeling, Institute of Geographic Sciences and Natural Resources Research, Chinese Academy of Sciences, Beijing 100091, China; wangjinsong@igsnrr.ac.cn
6   Institute of Biology/Geobotany and Botanical Garden, Martin Luther University Halle-Wittenberg, Am Kirchtor 1, 06108 Halle, Germany; helge.bruelheide@botanik.uni-halle.de (H.B.); stefan.trogisch@botanik.uni-halle.de (S.T.)
7   German Centre for Integrative Biodiversity Research (iDiv) Halle-Jena-Leipzig, Puschstr 4, 04103 Leipzig, Germany
*   Correspondence: bws2007@163.com; Tel.: +86-157-9769-0860
†   Han-Jiao Gu and Can-Can Zhang contributed equally to this work.
‡   Current address: No. 1101 Zhimin Road, Economic & Technological Development Zone, Nanchang 330045, China.

**Abstract:** Plant endemic species are the result of continuous evolution under the combined action of long-term climatic and geological conditions. There are 534 bamboo species in China, and 371 endemic species account for about 70% of all bamboo species. However, little is known about the differences in the rhizome evolution rate between endemic and non-endemic bamboos. Here, we collected morphological traits (height and leaf length) and environmental variables (including climate, space, and soil) of all 534 Chinese bamboo species to determine the relative contribution of environmental factors and traits of bamboo rhizome evolution with different endemism by structural equation modeling. We found that endemic bamboo had a higher speciation rate than non-endemic bamboo. The distribution centers of sympodial bamboos are mainly located in the mountain range of southwest China, while amphipodial and monopodial bamboos are distributed with higher latitude farther east in China. The height of non-endemic sympodial and monopodial bamboos was significantly higher than endemic sympodial and monopodial bamboos. The leaf length of non-endemic sympodial bamboos was significantly higher than endemic sympodial bamboo, while the leaf length of non-endemic amphipodial bamboo was significantly lower than endemic amphipodial bamboo. Environmental factors and traits explain 47% of the evolutionary variation of non-endemic bamboo species, while they explain 17% of that of endemic bamboo species. Longitude, latitude, and leaf length were the critical factors in the rhizome evolution of non-endemic bamboo, while longitude and height were the critical factors in the rhizome evolution of endemic bamboo. Our results imply that for higher species formation rates, endemic bamboo should be more concerned than non-endemic bamboo in the process of bamboo rhizome evolution. It will likely appear that new non-endemic bamboo species have a short leaf length in higher latitude and farther east in China (the lower Yangtze plain), as well as new endemic bamboo species with a low height farther east in China (the Wuyi Mountains). Meanwhile, ancient non-endemic bamboo with a long leaf length in Yunnan Province and ancient endemic bamboo with a high height in the Hengduan Mountains may be more

likely to become extinct. Our findings highlight the importance of understanding the difference in the rhizome evolution of endemic and non-endemic bamboos, which provides new insights into the conservation of Chinese bamboo biodiversity.

**Keywords:** bamboo; endemism; rhizome evolution rate; latitude; longitude; trait differentiation

## 1. Introduction

Since the publication of the book "Origin of Species" [1], the mechanism of speciation and the evolution of species have been widely studied. The phylogenetic and geographical origin of the Bambusoideae has always been a prominent topic in the study of Gramineae [2]. Bamboo is a vital plant resource with ecological, economical and cultural significance. Bamboos are mainly distributed in the tropical and sub-tropical zone, and a few in the temperate and frigid zone. China has the most abundant bamboo resources, and the geographical distribution of bamboos in China spreads from 40° N latitude to southern China [3]. The bamboo distribution regions are divided into the Yellow River and Yangtze River region, Yangtze River and Nanling Mountains region, south of China region and southwestern alpine region [3]. Yunnan Province is often regarded as the origin center of bamboo plants in China and one of the key origin centers of bamboo internationally [4]. As a long-life clonal plant, bamboo produces vegetative units (ramets) through vigorous rhizome growth [5]. The rhizome of bamboos can be divided into three divergent forms: the sympodial, amphipodial, and monopodial rhizome, according to the rhizome structure and bamboo pole clustering [3,6]. The proper sympodial rhizome is short and thick, usually curved upwards, and solid with asymmetrical internodes. Proper monopodial rhizomes are long and thin and have only short necks. Amphipodial rhizomes produce both typical pachymorph and typical leptomorph rhizomes in the same plant [3]. Sympodial rhizomes are likely ancestral within the tribe of Bambuseae [7], and monopodial is the advanced type [4]. The evolutionary order of bamboo vascular bundles is: double broken-waist type and broken-waist type with sympodial rhizomes, tight-waist type, and, finally, open-type and semi-open type bundles with monopodial rhizomes, which also reflect the evolution of bamboo [4,8,9]. Bamboo evolution might be best reflected by the emergence of different rhizome types [9]. The phylogeny of bamboo reflects the adaptive evolution of bamboos which adapt to the complex and changing environment to better acquiring water, nutrients, light, and space etc. [3,10]. Therefore, more attention should be paid to the research on the rhizome system when conducting comparative ecological research on bamboo.

Based on the existing evidence and the geographical distribution of the basal taxa, Bambusoideae probably originated from Gondwana in the Tertiary period and then differentiated into a group adapted to a forest habitat [11]. Endemic species resulted from the continuous occurrence and evolution of regional plant systems under the combined action of long-term climate and geological conditions. The distribution of endemic species is limited to specific areas, and so its distribution range is limited to a certain extent [3]. There are 371 endemic bamboo species in China, belonging to 23 genera and accounting for about 70% of all bamboo species in China [12]. Due to the different reasons for their formation, endemic species can be divided into epibiotic endemic species (paleoendemic species), neo-endemic species, and anthropogenic endemic species [13]. The ancient relict population reaches senescence after prime years, and no new endemic species appear anymore. Since Gramineae originated in the late Tertiary period, which is also the origin of the neo-endemic angiosperms [14], endemic bamboo can be considered as neo-endemic species. Rapid evolutionary change often takes place in small and isolated populations [4]. Neo-endemic species often derive from in situ diversification, and can arise through narrowly restricted species from rapid radiations [15,16]. As a kind of young subfamily, bamboos can show more diverse endemism [17]. Studying the endemism of different bamboo plants can help to better understand the origin, differentiation, and evolution of bamboo flora.

However, uncertainty remains regarding the evolutionary rate of bamboos with different endemism.

Species and their formed communities are the product of their evolutionary history and geographical environment on a large scale [18]. Ecological processes and evolutionary processes are the main driving forces for the formation of species' richness patterns. On a large spatial scale, geographic location determines the proportion of endemic genera in an area, and the heterogeneity of a habitat (or the complexity of topography) and climate factors also affects this [19]. Changes in topography have contributed to endemic species differentiation. Altitudinal gradients often cause high spatial variation in climate, soil, and hydrology, leading to genetic mutations or polyploid formation [17]. Climate change has a particularly significant impact on the richness of endemic species [20]. Furthermore, as a key driver of biodiversity patterns, edaphic variation influences the relative importance of speciation, dispersal, ecological drift, niche selection and their interactions [21]. In addition to environmental factors, variation in traits also affects the evolution of endemic and non-endemic species. The individuals of the same communities display trait differentiation under different environmental conditions to a certain extent, and the trait differences resulting from the geographical differences reflect the evolving ways and adapting strategies of the communities under different habitat conditions [22]. Non-endemic species can adapt to a broader range of environments and have a strong diffusion ability [3]. The trait differentiation in different environments can reduce the pressure from resource competition in the evolution of species [6]. Plant functional traits can be used to reveal the relationship between trait evolution and species evolution. Many plant functional traits are significantly affected by the history of phylogeny during their occurrence and development [23], which can enhance the explanatory power of speciation. Bamboo plants will also adopt morphological plasticity to respond to habitat changes, and trait differentiation is compatible with the status of habitat resources. Therefore, evaluating the relative contributions of various factors to the evolutionary processes of bamboo endemism is vital to understand the mechanism of Chinese bamboo speciation.

The growth and reproduction of bamboo are unique in that they have a large potential for growth and morphological plasticity [24]. As an important clonal plant, the genetic structure of bamboo is relatively stable [5] and bamboo does not have secondary growth. Although bamboos can reach the size of trees, the culms of woody bamboos can complete pole growth and grow into new bamboos in just a few months [3]. Moreover, the flowering and fruiting of bamboo is often a rare event occurring only every few decades, and some bamboo species even have no record of flowering and fruiting [5,25]. As the original type of bamboo, sympodial bamboos are mostly distributed in high heat and high humidity areas. They have large leaves and a tall culm height conducive to sufficient water supply during the shooting stage [4]. The more advanced types, such as amphipodial and monopodial bamboos, are the opposite. The shorter culm height and smaller leaves of these species make the amphipodial and monopodial bamboos more adaptable to the adverse environment during evolution [4]. Moreover, bamboos can intercept more light energy in the vertical direction with the increase in culm traits. They can also adapt to the environment-induced variation in light resources by adjusting the leaf length to obtain more light energy in the horizontal direction to drive the evolution and the endemic distribution of bamboos [12].

Due to the extraordinary growth and reproductive characteristics of bamboo, our study collected and investigated the distribution of bamboo in China and attempted to determine the relative contribution of environmental factors and traits to the rhizome evolution for all Chinese bamboo species by SEM. Structural equation modeling (SEM) enables scientists to use field data to test hypotheses about causal relations and pathways of bamboo rhizome evolution. As the response–effect framework has the potential to incorporate bamboo functional traits into the impacts of the environmental conditions on bamboo rhizome evolution, disentangling the complex interactions among environment, traits, and bamboo rhizome evolution, means that this method is well suited for the analysis

of SEM [26]. We hypothesized (H1) that the rhizome evolution rate of endemic bamboo species would be faster than that of non-endemic bamboo species due to a higher speciation rate. We also hypothesized (H2) that geographic isolation would play a critical role in the rhizome evolution of endemic bamboo species, while trait differentiation would play an essential role in the rhizome evolution of non-endemic bamboo species; because endemic species usually occur in geographically isolated areas, geographic isolation may block the gene exchange of bamboo and promote the rhizome evolution of new endemic bamboo species. However, different environmental adaptive evolution across the wide range of species distribution affects the genetic variation and trait differentiation of non-endemic bamboo. Our study attempts to evaluate how environmental factors and functional traits influence the bamboo rhizome evolution of different endemism and provide new insights into seeking endemic or nonendemic new bamboo species and optimizing the conservation strategy of Chinese bamboo resources.

## 2. Materials and Methods

### 2.1. Research Area

The study was carried out in China, one of the important distribution centers of bamboos globally. The bamboo forest area in China is 5.2 million ha, accounting for nearly 1/4 of the total area of bamboo forests globally. Bamboos usually grow in areas where the climate is warm and humid, and the soil is slightly acidic, including basins, mountains, hills and plateaus from 150 m to 3850 m asl [3].

### 2.2. Species Distribution Data

In this study, we considered bamboos naturally distributed in China because the changes in the distribution pattern of planted bamboos could be largely affected by human activities. The geographical distribution information of bamboos grown naturally in China was mainly collected from the *Flora of China* [27] and the Chinese Field Herbarium database (CFH, http://www.cfh.ac.cn/ accessed on 15 April 2020). A small part of the distribution data was obtained by our field surveys. For each bamboo species, we calculated the location of the distribution centers (i.e., latitude and longitude weighted by count) based on all distribution data [28]. Furthermore, we extracted the longitude and latitude of distribution centers for all bamboos to plot the distribution maps of bamboos in China with different rhizome types and endemism types by ArcGIS 10.2 (ESRI, Redlands, CA, USA).

### 2.3. Morphological Traits Data

All the Chinese bamboos, including 534 bamboo species from 34 genera, were studied (Table S1). According to the rhizome type, the bamboo species could be classified into 246 species of sympodial bamboo, 110 species of amphipodial bamboo, and 178 species of monopodial bamboo. Of the endemism types, 371 species were endemic to China and 163 were non-endemic species [12]. From our data of all bamboos of Chinese flora, we selected all bamboos distributed only in China based on *Flora of China* and China Biodiversity Red List: Higher Plants (http://www.mee.gov.cn/gkml/hbb/bgg/201309/t20130912_260061.htm accessed on 15 April 2020). In this study, the endemic data we collected included only indigenous species, which excluded all naturalized, exotic, and cultivated species.

Morphological traits, including potential culm height and leaf length, were selected for the study. Most of the trait values were collected from the *Flora of China*, while a small part of trait values was collected from our field investigations. We measured 20 individuals for each bamboo species sampled from field investigations. We took the maximum height value for the potential culm height and the mean value for leaf length. If the mean value of leaf length was missing, the mean of the maximum and minimum values of leaf length were used. Potential height was closely related to the growth form of species, the position of species in the vertical light gradient of vegetation, competitive activity, reproductive capacity, and the reproduction and potential life span of the whole plant [29]. Leaf length

was closely related to the leaf area and was the most common metric for leaf size, defined as the one-side or projected area of an individual leaf [29].

### 2.4. Environmental Data

Climate data were downloaded from WorldClim (http://www.worldclim.org/ accessed on 21 February 2020) based on the coordinates (longitude and latitude) of the distribution centers for each bamboo species. To avoid collinearity of climate variables, we excluded the highly relative climate variables by Pearson correlation coefficient r > 0.95. Then, we selected the climate factors which had higher coefficient correlat ions with height and leaf length, according to correlation analysis. (Table S2). Finally, we chose the annual mean temperature and mean temperature of the driest quarter as climatic factors.

The data for the soil type, soil texture, soil particle composition (sand fraction, silt fraction, and clay fraction), soil nutrients (organic matter, total nitrogen, total phosphorus, and total potassium), and soil pH of the bamboo, were gathered from the China Soil Database (http://vdb3.soil.csdb.cn/ accessed on 26 February 2020). To avoid the collinearity of soil variables, we excluded the highly relative soil variables by Pearson correlation coefficient r > 0.8. Then, we selected the soil factors which had higher coefficient correlations with height and leaf length, according to correlation analysis. (Table S3). Finally, clay fraction and total potassium were used as edaphic factors in this study.

### 2.5. Statistical Analysis

One-way ANOVAs were applied to explore the variations of environmental variables and trait indices across different rhizomes of non-endemic and endemic bamboos. Two-way ANOVA was used to explore the influence of different rhizomes, endemism, and their interactions on environmental factors and traits. Structural equation modeling (SEM) [30] was performed to explore how functional traits responded to environmental factors and affected the rhizome evolution of non-endemic and endemic bamboos. Considering the rhizome evolution, we quantified the most primitive sympodial bamboo as 1, amphipodial bamboo was quantified as 2, and the most advanced monopodial bamboo was quantified as 3 in SEM. Latitude and longitude were set as exogenous variables; Clay fraction, Total potassium, Annual mean temperature, and Mean temperature of driest quarter were considered endogenous variables. All predictors were standardized to have a mean of 0 and a standard deviation of 1 to improve the interpretability of regression coefficients [31]. Maximum likelihood estimation was used to test whether the data fit the model. A total standardized effect was applied to determine the relative importance of environmental factors and traits on the rhizome evolution of bamboos with different endemism and to identify the key drivers which affected the rhizome evolution processes of non-endemic and endemic bamboos. The distribution patterns of different rhizome or endemism types of bamboos were mapped by ArcGIS 10.2. All statistical analyses were completed by the basic and "lavaan" package in R 3.5.1 software [32].

## 3. Results

### 3.1. Distribution Patterns of Different Bamboos

In China, there are more endemic bamboos than non-endemic bamboos for each rhizome type bamboo (Figure 1A). The quantity of sympodial bamboo was the highest, followed by monopodial bamboos, and the quantity of amphipodial bamboos was the lowest for each endemic type of bamboo. The proportion of non-endemic sympodial bamboos (55%) was higher than that of endemic sympodial bamboos (42%); the proportion of non-endemic amphipodial (16%) and monopodial bamboos (29%) was lower than that of endemic amphipodial bamboos (23%) and monopodial bamboos (35%) (Figure 1B). The proportion of bamboos with different rhizome types decreased first and then increased for each endemic type of bamboo.

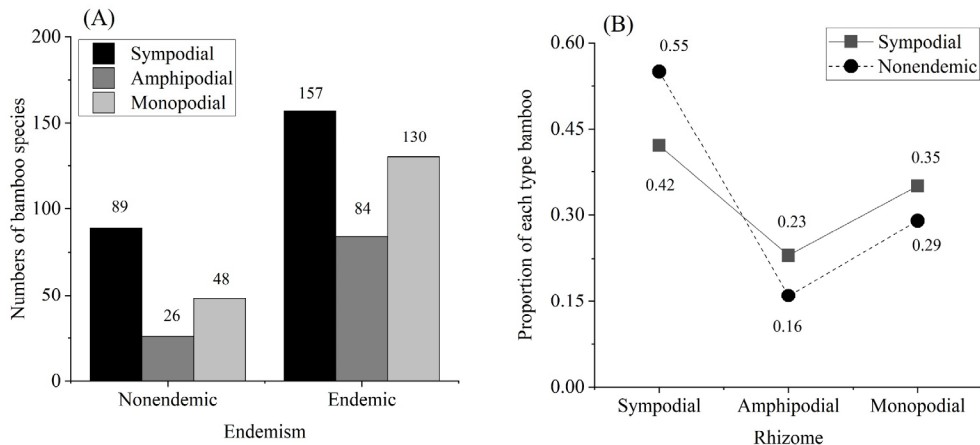

**Figure 1.** (**A**) Number of bamboo species; (**B**) Proportion of each type bamboo.

Bamboos in China are mostly distributed south of 30° N latitude, in southwest and south China (Figure 2). The distribution centers of non-endemic sympodial bamboos are mainly located in the Hengduan Mountains, the alpine gorge region of western Sichuan, Yunnan-Guizhou Plateau Guangdong and Guangxi Hills. The distribution centers of non-endemic amphipodial bamboos are located in the Hengduan Mountains, the alpine gorge region of western Sichuan, Yunnan-Guizhou Plateau, Wuling Mountains, Daba Mountains, Nanling Mountains, Guangdong and Guangxi Hills, the Middle-lower Yangtze Plain, Wuyi Mountains, and Taiwan Island, dispersedly. The distribution centers of non-endemic monopodial bamboos are mainly located in Nanling Mountains, Guangdong and Guangxi Hills, and Wuyi Mountains (Figure 2a). The distribution centers of endemic sympodial bamboos in China are mainly in the alpine gorge region of western Sichuan, Yunnan-Guizhou Plateau, Wuling Mountains, Guangdong and Guangxi Hills, and Nanling Mountains. The distribution centers of endemic amphipodial bamboos in China are mainly located in the alpine gorge region of western Sichuan, Wuling Mountains, Qingling and Daba Mountains, Nanling Mountains, Guangdong and Guangxi Hills, and Wuyi Mountains. The distribution centers of endemic monopodial bamboos are mainly located in the Hengduan Mountains, the alpine gorge region of western Sichuan, Yunnan-Guizhou Plateau, Nanling Mountains, Guangdong, and Guangxi Hills, and Wuyi Mountains (Figure 2b).

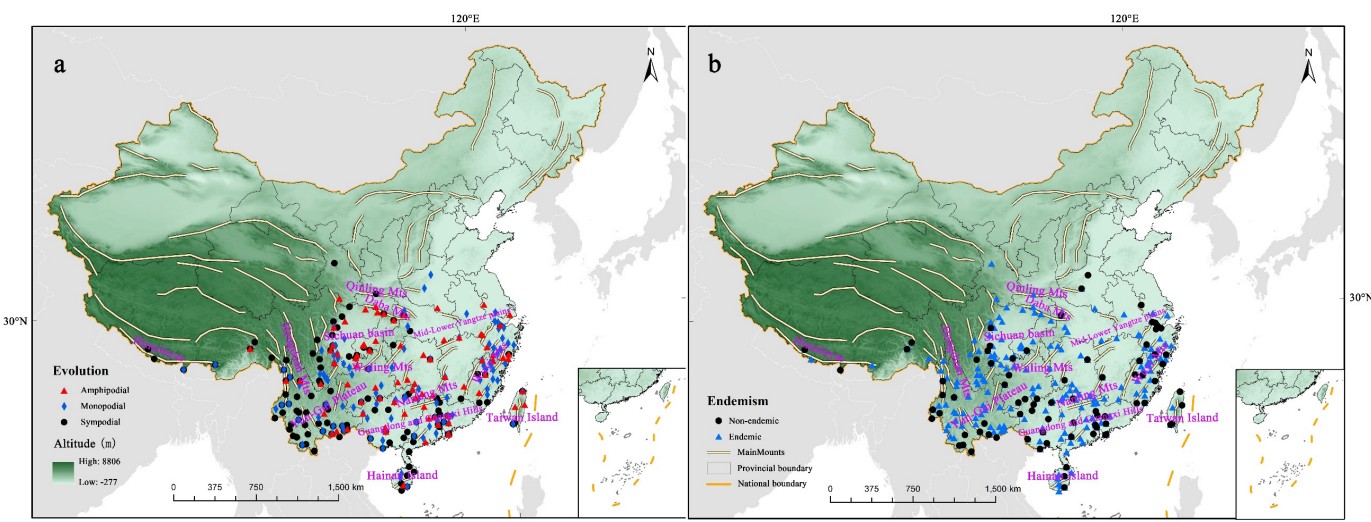

**Figure 2.** Distribution centers of bamboos. (**a**) Non-endemic bamboos; (**b**) Endemic bamboos.

The longitude of non-endemic monopodial bamboos was significantly farther east than that of endemic monopodial bamboos, but the longitude of sympodial and amphipodial

bamboos was not significantly different between different endemism types (Figure 3A). In terms of non-endemic bamboos, the longitude of the amphipodial bamboo was significantly farther west than that of monopodial bamboos, whereas it was significantly farther east than that of sympodial bamboos. Endemic sympodial bamboos located farther west in China, were significantly different from endemic amphipodial and monopodial bamboos. The latitude of non-endemic sympodial bamboos was significantly lower than that of endemic sympodial bamboos, but the latitude of non-endemic monopodial bamboos was significantly higher than that of endemic monopodial bamboos (Figure 3B). The latitude of sympodial bamboos was lower than that of amphipodial and monopodial bamboos, but there was no significant difference between amphipodial and monopodial bamboos, regardless of endemism types. Additionally, the interaction between rhizome types and endemism had significant effects on longitude and latitude ($p < 0.05$).

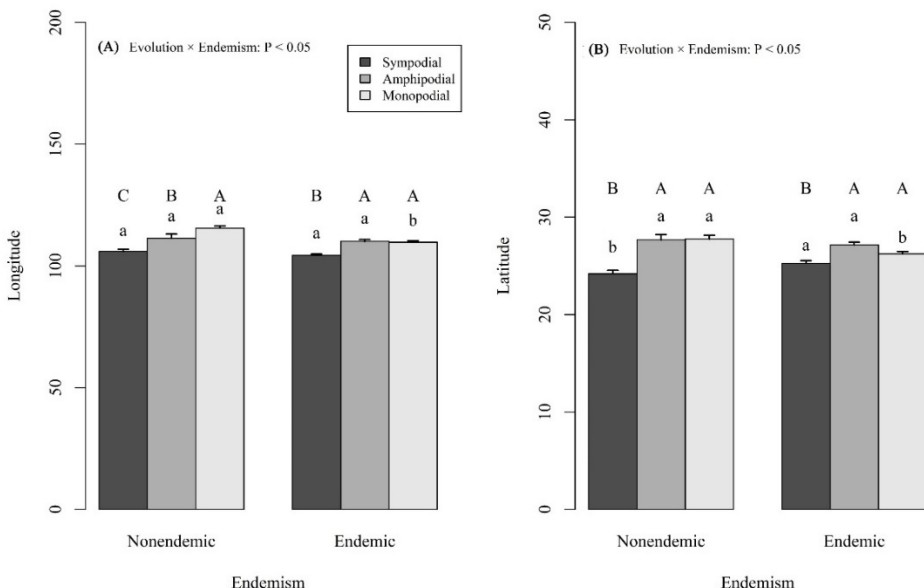

**Figure 3.** Variations of the longitude and latitude between different endemism and among different rhizome types. Boxes with different letters differ significantly at $p < 0.05$. The lowercase letters mean the difference between different endemism types. The uppercase letters mean the differences among different rhizome types. "×" means the interaction between two factors.

### 3.2. Variations in Functional Traits across Different Bamboos

For sympodial and monopodial bamboos, the height of non-endemic bamboos was significantly higher than that of endemic bamboos, but the height of amphipodial bamboos was not significantly different between different endemism (Figure 4A). The height of monopodial bamboos was lower than that of sympodial bamboos, and higher than that of amphipodial bamboos, regardless of the endemism. The leaf length of non-endemic sympodial bamboos was higher than that of endemic sympodial bamboos, whereas that of non-endemic amphipodial bamboos was significantly lower than that of endemic amphipodial bamboos (Figure 4B). For non-endemic bamboos, the leaf length of sympodial bamboos was higher than that of monopodial bamboos, while leaf length of endemic amphipodial bamboos was higher than that of endemic sympodial and monopodial bamboos. The interaction between rhizome types and endemism had significant effects on height and leaf length ($p < 0.05$).

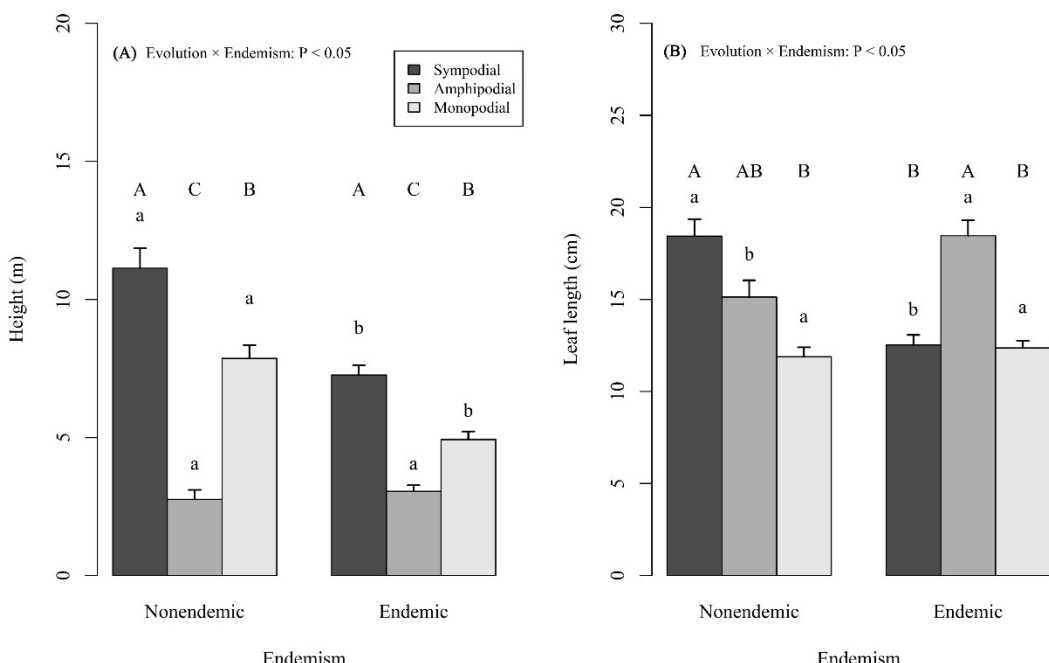

**Figure 4.** Effect of endemism on traits of different rhizome types. Boxes with different letters differ significantly at *p* < 0.05. The lowercase letters mean the difference between different endemism types. The uppercase letters mean the differences among different rhizome types. "×" means the interaction between two factors.

### 3.3. How Environmental Factors and Traits Influence Rhizome Evolution of Bamboos with Different Endemism

In terms of non-endemic bamboos, the longitude and latitude directly and positively influenced the rhizome evolution process (Figure 5A). The change in soil factors was directly and negatively related to the change of longitude. Longitude significantly increased the total potassium levels but decreased soil clay fraction, further decreased height, and, finally, increased rhizome evolution. Moreover, longitude significantly decreased soil factors and decreased leaf length, further decreased height, and, finally, influenced the rhizome evolution of non-endemic bamboos. The change in climatic factors was directly and negatively related to latitude and positively related to longitude. With the increase in longitude and the decrease in latitude, both the annual mean temperature and mean temperature of the driest quarter increased significantly, further increased leaf length and height, and thereby negatively influenced the rhizome evolution of non-endemic bamboos.

In terms of endemic bamboos, the longitude and latitude also directly and positively influenced the rhizome evolution process (Figure 5B). The changes in soil factors were directly and negatively related to the change of longitude. A higher longitude significantly increased total potassium levels but decreased soil clay fraction, further decreased height, and, finally, increased the rhizome evolution of endemic bamboos. Higher longitude increased leaf length and culm height and thereby negatively influenced the rhizome evolution process of non-endemic bamboos. Moreover, with the increase in longitude and the decrease in latitude, both the annual mean temperature and the mean temperature of the driest quarter increased significantly, further increased leaf length and height, and thereby negatively influenced the rhizome evolution of endemic bamboos. The higher annual mean temperature and mean temperature of the driest quarter in the climate factors also increased soil clay fraction but decreased the total potassium levels, which increased height, and thus negatively influenced the rhizome evolution of endemic bamboos. However, there was no direct effect of leaf length on the endemism process of bamboos.

(A)                                                                                                 (B)

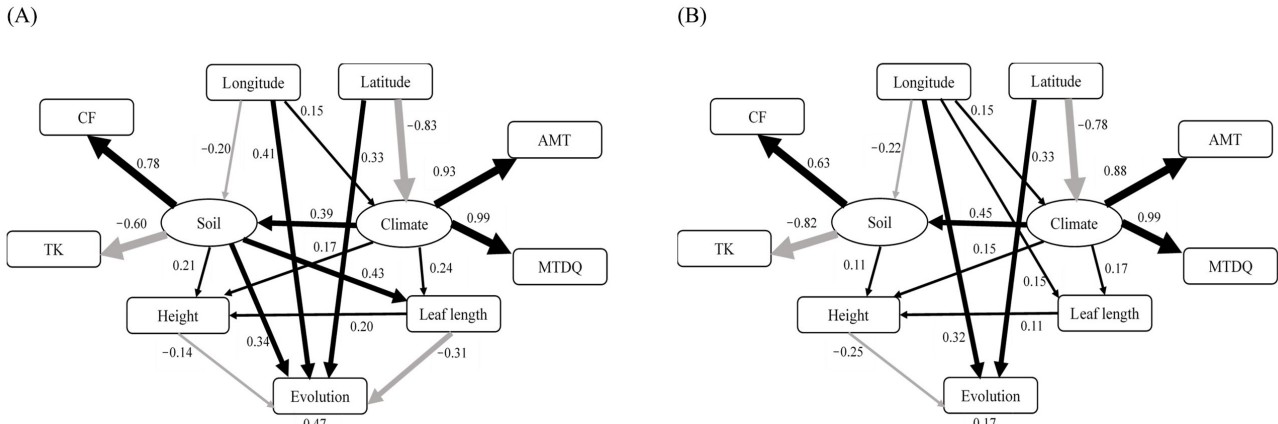

**Figure 5.** The effect of structural equation models (SEM) of environmental factors on trait variations during evolution of non-endemic bamboos (**A**) ($X^2$ = 28.401; *df* = 17, *p* = 0.056; *RMSEA* = 0.068; *AIC* = 84.859 in SEM-A) and endemic bamboos (**B**) ($X^2$ = 13.636; *df* = 17, *p* = 0.050; *RMSEA* = 0.043; *AIC* = 84.401 in SEM-B) in Chinese bamboos. Note: Black and gray arrows indicate significant positive and negative relationships (*p* < 0.05), respectively. Arrow width denotes the strength of the causal influence and the numbers are standardized path coefficients. The percentage of explanation of the models on the chosen factors is represented by $R^2$. CF, Clay fraction; TK, Total potassium; AMT, Annual mean temperature; MTDQ, Mean temperature of driest quarter.

The SEM of non-endemic bamboos explained 47% of the variation of rhizome evolution, while the SEM of endemic bamboos explained 17% of the variation of rhizome evolution (Figure 5). The standardized total effects (including direct and indirect effects) indicated that longitude and latitude had positive effects, while height, leaf length, and climate had negative effects on the rhizome evolution of both endemic and non-endemic bamboos (Figure 6). Soil factors had a positive effect on the rhizome evolution of non-endemic bamboos but had a negative effect on the rhizome evolution of endemic bamboos. In short, longitude, latitude, and leaf length were the key factors in the rhizome evolution of non-endemic bamboos, while longitude and height were the key factors in the rhizome evolution of endemic bamboos (Absolute value of standardized total effect > 0.2).

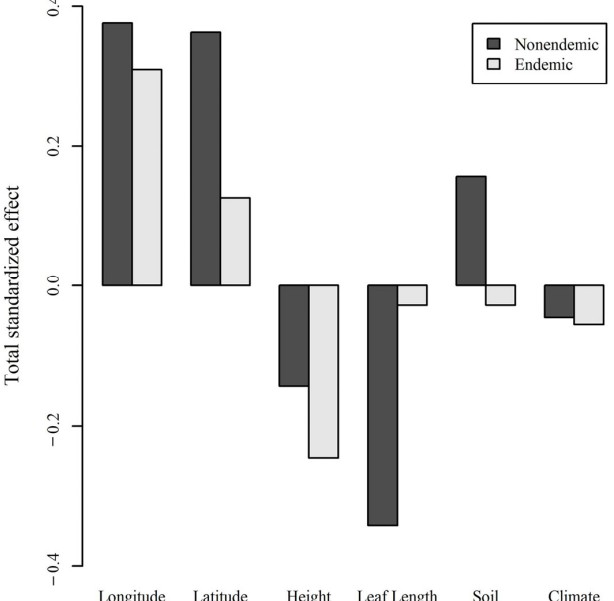

**Figure 6.** Total standardized effect of factors on the evolution process of bamboos with different endemism.

## 4. Discussion

We collected morphological traits (height and leaf length) and environmental variables (including climate, space, and soil) of all 534 Chinese bamboos to determine the relative contribution of environmental factors and traits of bamboo rhizome evolution with different endemism by SEM. We obtained two key findings. First, endemic bamboos had a higher speciation rate than non-endemic bamboos. Second, the rhizome evolution of endemic and non-endemic bamboo species was driven by different factors. Longitude, latitude, and leaf length were the key factors in the rhizome evolution of non-endemic bamboos, while longitude and height were the key factors in the rhizome evolution of endemic bamboos.

### 4.1. Distribution Patterns of Bamboos with Different Endemism

As expected, our results indicated that the species formation rate of endemic bamboos was higher than that of non-endemic bamboos. There were more endemic bamboo species than non-endemic bamboo species in each rhizome type. Moreover, the proportion of non-endemic sympodial bamboos was higher than that of endemic sympodial bamboos; the proportion of non-endemic amphipodial and monopodial bamboos was lower than that of endemic amphipodial and monopodial bamboos. These results demonstrated that the species formation rates of endemic amphipodial and monopodial bamboos were higher than that of non-endemic amphipodial and monopodial bamboos. Some studies also found that the formation rate of endemic species was higher than that of non-endemic species [33]. The Poaceae originated in the Late Cretaceous (c. 96 Ma), as shown by three plastid markers and four fossil calibration points, and the diversification of bamboos occurred during the Miocene period (23.7 to 5.3 million years ago) as documented by macrofossils or pollen fossil records [34]. In the time scale of the ecological process, the population may rapidly undergo adaptive evolution to the environment and competition by natural selection [35]. Furthermore, geographic isolation is also one of the causes of endemic species. Many species originated through allopatric divergence in geographically isolated populations of the same ancestral species, and allopatric speciation can occur rapidly [36]. Compared with narrowly distributed endemic species, non-endemic species are less likely to threaten extinction [37]. Endemic bamboos generally have a stronger asexual reproduction and viability in a narrow and complex environment due to physiological integration and phenotypic plasticity [38], conducive to its rapid evolution. Based on this, we can speculate that environmental constraints will intensify the competition for resources of endemic bamboo, thus promoting the rhizome evolution rate.

Furthermore, we found that the distribution centers of endemic and non-endemic sympodial bamboos are mainly distributed in the mountainous range of south and southwestern China with a low latitude and longitude. In contrast, amphipodial and monopodial bamboos are distributed in mountainous ranges with a higher latitude and longitude. In China, bamboos are mainly distributed south of the Qinling-Huaihe River Basin, with a wide range of natural distribution [3]. Compared with amphipodial bamboo and monopodial bamboo, sympodial bamboo is well known to have a higher demand for temperature and precipitation [39]. The southwestern region with a low latitude and longitude is affected by the Indian Ocean monsoon and has more suitable rainfall and temperature conditions for sympodial bamboo species [40]. The rich precipitation and heat conditions in tropical and southern subtropical regions are also suitable for bamboo growth, especially sympodial bamboos. In contrast, amphipodial and monopodial bamboos are more resistant to lower temperatures and drought because of the higher rooting depths of their rhizomes [39]. Moreover, their requirements for soil conditions are less than those of sympodial bamboos, and they also show higher adaptability resulting in a broader distribution than amphipodial and monopodial bamboos.

The environmental effects of endemic phenomena include climatic conditions, geomorphic factors, soil factors, and edge effects in modern and geological periods [41]. Our findings showed that the distribution centers of endemic bamboos in the alpine gorge regions of western Sichuan, Qinling-Daba Mountain, Hengduan Mountains, Nanling

Mountains, Wuyi Mountains, and Yunnan-Guizhou Plateau are more abundant than that of non-endemic species. The distribution centers of the paleoendemic genera are mainly in eastern China, and the neo-endemic genera are mainly distributed in the west and southwest of China [42]. The geological history of eastern China may be relatively stable, and is home tp the paleoendemic species centers of China [41]. The Qinling Mountain is the natural boundary between the north and the south of China and is a central biodiversity area in China. The unique geographical location and environment are conducive to forming endemic species [13]. Furthermore, it is generally assumed that the Hengduan Mountains are one of the areas of endemism and have the largest proportion of the Chinese endemic genera in China [43]. The heterogeneous habitat and geographic isolation in the Hengduan Mountains are also important driving forces for the rapid diversification of alpine bamboos [44]. During the formation of the Hengduan Mountains, the endemic species began to adapt to the cold and changeable climate of the mountains as the ground continued to rise, resulting in adaptive traits at high altitudes. The alpine gorge region of western Sichuan is also located in the northern part of the Hengduan Mountains, where the terrain changes dramatically. It has a large number of species and preserves a large number of ancient and newly differentiated endemic species [13]. Therefore, combined with complex terrain, it is easier to form unique habitats conducive to the distribution and growth of endemic species in China. The increase in altitude will increase the degree of geographic isolation, which is also conducive to the distribution of bamboos in the process of endemism.

### 4.2. Variations in Functional Traits of Bamboos with Different Functional Groups

Our research found that in the same endemism type, bamboos with different rhizome types had significnat differences in height and leaf length. The growth height of sympodial bamboos was the highest, followed by monopodial bamboos, while amphipodial bamboos showed the lowest growth height in each endemic type. Plant strategies to respond to the environment depend on their functional trait values [23]. Bamboos can adapt to different resource levels through morphological shaping under different levels of soil nutrients, soil moisture, and light resources [38]. Sympodial bamboos have high requirements for water and heat conditions, and higher individuals will appear in the environment with sufficient water and heat [39,45]. Sympodial bamboos with no horizontal bamboo rhizome form dense clumps because they cannot spread and grow underground for long distances [39]. The plants are clustered in a particular space, which has a specific restriction on the absorption, synthesis, and storage of nutrients, as well as the absorption of light energy [46]. The larger height of sympodial bamboos is also conducive to the vertical absorption of light energy and promotes growth. Our results also show that the endemic amphipodial bamboos have the maximum leaf length. Leaf size can reflect the horizontal interception of light energy and the ability to capture carbon [47]. After the amphipodial bamboo shoots out, it takes the shortest time for the branches and leaves to spread and for the completion of growth in culm height [39]. From the perspective of endemism, we found that in the same rhizome type, bamboos with different endemism types have significant differences in height and leaf length. In China, the height of endemic sympodial bamboos and monopodial bamboos are significantly smaller than that of non-endemic sympodial and monopodial bamboos. The leaf length of endemic sympodial bamboos is also significantly smaller than that of non-endemic sympodial bamboos. A study found that endemic plant species distributed in narrow ranges may be more adaptable to particular habitats and form a series of adaptive traits, such as lower height and specific leaf area [48]. Non-endemic species allocate more energy to reproduction and growth, thus following the acquisitive strategy with a higher photosynthetic rate and a larger specific leaf area, as well as rapid growth and low wood density of stems [23].

Overall, our results indicated that sympodial bamboos have the highest growth height and the shortest leaf length of endemic bamboos, whereas amphipodial bamboos have the lowest height and the longest leaf length. There seems to be a tradeoff between horizontal

and vertical interception of light energy through the plasticity of leaf length and height in endemic bamboos. The trade-off relationship between traits is a combination of traits after natural selection, also known as "ecological strategy" [23]. The investment in height is conducive to improving the access to the vertical light resource [49]. Larger bamboos can obtain more energy in the vertical direction with the elongation of internodes under sufficient light conditions after shooting, thereby reducing the horizontal competition for light resources. Shorter plants or individuals have a disadvantage over higher plants in that they can only intercept less vertical light [49]. The shortest endemic amphipodial bamboos can invest less energy in the culm and more carbon input in the long leaf to capture the horizontal light.

### 4.3. Factors Influencing the Rhizome Evolution of Bamboos with Different Endemism

The result of SEM also showed that environmental factors and traits explained 47% of the variation in non-endemic bamboo evolution, which was higher than that of endemic bamboo species. In the long term, environmental changes and climate fluctuations may lead to different geographical environments, which cause the niche differentiation of the population. Furthermore, niche differentiation leads to the evolution of sexual selection or differentiation, which induces phenotypic diversification of ancestral populations and finally produces species differentiation [50]. A review of 54 published studies also concluded that correlations between traits and rarity strongly depend on the pant phytogeographic and ecological context [51]. Non-endemic species are widely distributed and have stronger resistance and adaptability to environmental changes. Studies on *Quercus* have found that species with larger distribution ranges generally have significantly higher genetic diversity than endemic species, and there may be a variety of phenotypic changes to cope with different growth environments [52]. Our results indicate that niche differentiation caused by environmental change has an important effect on non-endemic bamboo species.

Moreover, in our analyses, we found significant differences in environmental factors between different endemism and different rhizome types. Longitude, latitude, and leaf length were the most crucial factors influencing the rhizome evolution of non-endemic bamboos (Figure 5A). Environmental diversity and competition for resources affect the speed of species evolution [53]. China spans 6000 km from east to west [54]. The longitude affects the location of the land, sea, and topography, which also affects the changes in rainfall and temperature. In China, the topography gradually increases from east to west. Changes in altitude lead to a decrease in temperature and the formation of a continuous temperature gradient. The precipitation pattern also significantly correlates with longitude and gradually decreases from southeast to northwest, forming an obvious precipitation gradient [55]. Additionally, changes in latitude directly impact temperature because the solar irradiation angle decreases, heat decreases with latitude, and the temperature becomes colder. Differences in temperature and precipitation can also lead to different traits produced by natural selection in different populations [23]. Leaf traits have the highly adaptable and self-regulatory capacity to complex environmental conditions. By adjusting the metabolic activity for the allocation of plants, climate directly affects the morphology of the leaves [23]. Furthermore, the longitude and latitude has a positive effect, whereas leaf length has a negative effect on the rhizome evolution of non-endemic bamboos. These results imply that the fastest-evolving non-endemic bamboo should have a short leaf length and occur in northeast China, whereas ancient non-endemic bamboo should have a long leaf length and occur in southwest China. According to the distribution center of non-endemic bamboo (Figure 2a), we deduce that the new non-endemic bamboo species with a short leaf length will likely appear in the lower Yangtze plain, while ancient non-endemic bamboos with a long leaf length may become extinct in the Yunnan Province.

According to our hypothesis, SEM showed that longitude and height played an important role in the rhizome evolution of endemic bamboos (Figure 5B). Due to the influence of certain geological or climatic factors, ancestral populations have formed several small populations that are geographically isolated and separated from each other. There is

no continuous gene flow among these isolated populations, leading to genetic divergence and differentiation into different species [56]. In endemic phenomena, species formation is related to long-term evolution and geographical differentiation, and environmental changes are more sensitive to the formation of endemic species [19]. With the change of longitude from east to west, China's terrain gradually increases, and the precipitation and monsoon environment also gradually change. Uplifts and the rise of altitude can form a series of complex topography, climate, and habitats, leading to population isolation on high peaks and ridges [44] and the rapid promotion of species diversity and endemism formation. Furthermore, longitude has a positive effect, whereas height has a negative effect, on the rhizome evolution of endemic bamboos. These results hint that the fastest-evolving endemic bamboo should have a low height and occur in eastern China, whereas ancient endemic bamboo should have a high height and occur in western China. According to the distribution center of endemic bamboo (Figure 2b), we infer that new endemic bamboo species with a low height will likely appear in the Wuyi Mountains, while ancient endemic bamboos with a high height may be more likely to become extinct in the Hengduan Mountains. Recently, two new endemic bamboo species with a short culm height were found near the Wuyi Mountains [57,58].

## 5. Conclusions

In conclusion, we found that the rate of rhizome evolution of endemic bamboos was faster than that of non-endemic bamboos, which was verified by the higher speciation rate of endemic species. This result implied that endemic bamboo should be more significant than non-endemic bamboo for exploring bamboo rhizome evolution. Additionally, longitude, latitude, and leaf length were the key factors in the rhizome evolution of non-endemic bamboos, while longitude and height were the key factors in the rhizome evolution of endemic bamboos. Based on the distribution center of bamboos, new non-endemic bamboo species with short leaf lengths will likely appear in the lower Yangtze plain, while ancient non-endemic bamboo with long leaf length may be more likely to become extinct in Yunnan Province. Furthermore, new endemic bamboo species with low height will likely appear in the Wuyi Mountains, while ancient endemic bamboos with high height may be more likely to become extinct in the Hengduan Mountains. Therefore, we suggest that new nature reserves should be constructed to protect the habitat of bamboo species with short leaf lengths in the lower Yangtze plain and with low height in the Wuyi Mountains. Moreover, we should also pay attention to or enact ex situ conservation on the ancient non-endemic bamboos with long leaf lengths in Yunnan Province and the ancient endemic bamboos with high height in the Hengduan Mountains, in order to prevent their extinction. Our findings highlight the importance of understanding the difference between the rhizome evolution of endemic and non-endemic bamboos, which provides new insights for the conservation of Chinese bamboo biodiversity.

**Supplementary Materials:** The following are available online at https://www.mdpi.com/article/10.3390/f12091280/s1, Table S1. The list of bamboo species in China. Table S2. Correlations among functional traits of bamboos and climatic factors. * indicates $p < 0.05$, ** indicates $p < 0.01$, *** indicates $p < 0.001$. AMT, Annual average temperature MDR, Mean diurnal range; IS, Isothermality; MTWM, Max temperature of warmest month; TAR, Temperature annual range; MTDQ, Mean temperature of driest quarter; AP, Annual precipitation; PDQ, Precipitation of driest quarter, Table S3. Correlations among functional traits of bamboos and soil factors. * indicates $p < 0.05$, ** indicates $p < 0.01$, *** indicates $p < 0.001$.

**Author Contributions:** H.-J.G., C.-C.Z. and W.-S.B. developed the original idea of the analyses presented in the manuscript; F.-S.C. and W.-S.B. designed the field and laboratory experiment; X.-M.F. and J.-J.L. helped with field sampling and laboratory measurement; J.-H.H. and J.-S.W. helped with the use of GIS; H.-J.G. and C.-C.Z. collected all data, performed all the statistical analyses and graphs, and wrote the first draft supported by H.B., S.T. and W.-S.B.; H.-J.G. and C.-C.Z. contributed equally to this work and share first authorship. All the authors contributed substantially to the revisions of the manuscript. All authors have read and agreed to the published version of the manuscript.

**Funding:** This work was supported by the National Natural Science Foundation of China (31760134) and Jiangxi Provincial Department of Science and Technology (20202ACBL215005).

**Data Availability Statement:** The data presented in this study are available in Supplementary Material Table S1.

**Acknowledgments:** We thank Li-Xin Tian, Peng-Yu Ma, Xue-Wen Shi for their help with field sampling and laboratory measurement. We thank the National Earth System Science Data Center in the Institute of Soil Science, Chinese Academy of Sciences for collating soil data. Gratitude is extended to Jin-Long Zhang, Curator with the Specimen Hall of Kadoorie Farm and Botanical Gardens (KFBG) for his aid in data processing and analyses.

**Conflicts of Interest:** The authors declare no conflict of interest.

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
