# Peer review of "The Bamboo Rhizome Evolution in China Is Driven by Geographical Isolation and Trait Differentiation"

_forests, doi:10.3390/f12091280_

Round 1
Reviewer 1 Report
This ms made extensive analyses of 500 some bamboo accessions in China for environmental and locational effects. It should be interesting to the readership of this journal.
Given that the data may be used by other researchers, it is important that the authors include supplemental datasets of the raw data of their study. Importantly, those raw dataset should include name or name for all 500 some accessions analyzed, with both the standard English or Pingin spelling as well as the Chinese characters or accession numbers if the Chinese character is absent
Reviewer 2 Report
This paper surveyed the occurrences of bamboo plants in China and tried to infer the bamboo rhizome evolution based on the correlation between geographical distribution, traits, and climatic factors. Since bamboo owns such a unique reproduction system and great economic value, evolutionary history research on bamboo is of great importance to community. I think that the numbers of samples are large, and authors analyzed data appropriately and presented data well. So, this study would have good contribution to deepening our understanding on ecology and evolution of bamboo.
However, I do not recommend publication of this study. In my feeling, local adaption and ecological niche factor analysis are necessary for inferring driven factors underlying ecology and evolution. This study solely described the relationship between different factors and traits, and infers causes shaping the pattern by performing SEM analyses. Although such findings would add our knowledge related to environmental factors for adaptive divergence of bamboo plants, the present study could not clearly explain how bamboo species adapt to the environments in which they occur, or how growth of bamboo population was affected by climate variables.
In addition to these weaknesses, authors need to revise English writing. Sentences within paragraphs need to be restructured, so the main ideas can be clear. Please don't describe figures in detail in the main text.
Major revision:
- The introduction section needs to be rewritten. Authors introduced broad aspects of bamboo, from phylogeny, origin, to physiology and reproduction, but the geographical distribution and trait differentiation, the main topic of this paper, seem to be underrated. I suggest the authors focus on these two aspects, and simplify the introduction of the other aspects. Additionally, what’s the main contribution/function of this research to community? It should be clearly described in the last paragraph of the introduction section. You may also want to add introduction of SEM, which is the main tool that you use.
- I suggest authors seek professional English editing service to improve the quality of the manuscript. Sentences don’t flow and it is hard for readers to grasp the main idea of each paragraph.
- I am not sure of retaining only the highly correlated climatic/soil factors for analyses. How much trait variation can be explained by these highly correlated factors? How about those less correlated factors? Also, why different r values (r > 0.95 for climatic factors, and r > 0.8 for edaphic factors) were used? How about adding topographical factors like elevation for analyses? As many variables were not analyzed, the discussion about precipitation and mountain landscape sounds irrelevant.
Minor revision:
Abstract
Readers like myself are not familiar with words of sympodial, amphipodial, and monopodial. Do they describe the subspecies or ecotypes of bamboo? Please clarify or you may want to reword.
Line28: Can you rephrase this sentence, please? It is hard to understand.
Line30: bamboos bamboo species? I feel bamboo is a general word, you may want to be more specific here.
Line33: bamboos bamboo plants?
Line37: mountainous range mountain range
Line36-38: Not clear. What do you mean by “low latitude and longitude”? I think readers are more used to “eastern” or “western” instead of “high/low longitude”.
Line38-41: Please rephrase these sentences.
Line46: Can you reword this sentence, please?
Line47-49: Not clear. Please rephrase this sentence.
Line52: for into
Line59-61: Weird. Does “species” particularly mean “plant species” here?
Line 65: “can be produced first”? I don’t understand. Please clarify.
Line68: Does “ca.” mean “nearly?
Materials and Methods
For 2.1 & 2.2, the description about sampling locations is not clear. Can you show a map and highlight the sampling/survey locations, please?
Line185: “We collected about 20 measurements got for each species.” I don’t understand what you mean.
Line188-192: These statements sound like results instead of methods.
Line264: Is it common to use “high” and “low” to describe “longitude”? To me, “eastern” and “western” sound more straightforward.
Results:
Line230-238: Can you show a table to state these numbers, please?
Line239-263: Can you combine and simply these two paragraphs? It is hard to catch what you mean.
Line 264-295: As the figures 3 & 4 have shown the comparison, it is repetitive to describe the differences in detail in the main text. Just highlighting a few important points would be enough.
Line297: Please rephrase this sentence.
Line322: I don’t understand what “evolution” represents here. How did you quantify “evolution” here? Please clarify.
Discussion
Can you put a summary paragraph at the beginning of the Discussion section, please?
Line333: I don’t think species biodiversity equals to species formation rate. For inferring species formation rate, evidence based on molecular clock analyses and fossil records are required.
Line361-366: As you excluded precipitation factors in your analyses, the discussion about precipitation sounds weak here.
Line368-389: Again, it is a pity that you did not include topographical factors in the SEM analyses, the discussion related to mountains sounds weak.
Line392-426: It is very hard to follow these two paragraphs. It would be great if you can put the main points in the first sentence, and expand your argument around the main points.
